# Stabilization of $N_6$ and $N_8$ anionic units and 2D polynitrogen layers in high-pressure scandium polynitrides

Andrey Aslandukov [1,2] ✉, Alena Aslandukova [1], Dominique Laniel[3], Saiana Khandarkhaeva[1], Yuqing Yin [2,4], Fariia I. Akbar [1], Stella Chariton [5], Vitali Prakapenka [5], Eleanor Lawrence Bright[6], Carlotta Giacobbe[6], Jonathan Wright [6], Davide Comboni[6], Michael Hanfland[6], Natalia Dubrovinskaia [2,4] & Leonid Dubrovinsky [1]

Nitrogen catenation under high pressure leads to the formation of polynitrogen compounds with potentially unique properties. The exploration of the entire spectrum of poly- and oligo-nitrogen moieties is still in its earliest stages. Here, we report on four novel scandium nitrides, $Sc_2N_6$, $Sc_2N_8$, $ScN_5$, and $Sc_4N_3$, synthesized by direct reaction between yttrium and nitrogen at 78-125 GPa and 2500 K in laser-heated diamond anvil cells. High-pressure synchrotron single-crystal X-ray diffraction reveals that in the crystal structures of the nitrogen-rich $Sc_2N_6$, $Sc_2N_8$, and $ScN_5$ phases nitrogen is catenated forming previously unknown $N_6^{6-}$ and $N_8^{6-}$ units and $_\infty^2(N_5^{3-})$ anionic corrugated 2D-polynitrogen layers consisting of fused $N_{12}$ rings. Density functional theory calculations, confirming the dynamical stability of the synthesized compounds, show that $Sc_2N_6$ and $Sc_2N_8$ possess an anion-driven metallicity, while $ScN_5$ is an indirect semiconductor. $Sc_2N_6$, $Sc_2N_8$, and $ScN_5$ solids are promising high-energy-density materials with calculated volumetric energy density, detonation velocity, and detonation pressure higher than those of TNT.

The discovery of nitrogen polymerization under high pressures has significantly extended the nitrogen chemistry. While the polymeric single-bonded nitrogen allotropes are formed only at pressures above 110 GPa[1–3], the introduction of electropositive elements facilitates breaking the $N_2$ triple-bond and initiates nitrogen catenation at significantly lower pressures. Indeed, under high-pressure high-temperature conditions nitrogen easily reacts with metals and forms numerous compounds featuring charged nitrogen $N_2^{x-}$ dimers[4–19] at low-to-mild pressures (5-50 GPa), or various catenated nitrogen units (e.g. tetranitrogen $N_4^{4-}$ units[20], pentazolate $N_5^-$ rings[21–23], $N_6$ rings[24–26], and $N_{18}$ macrocycle[27]) and 1D-polynitrogen chains[20,28–33] at mild-to-high

pressures (>50 GPa). Some of the nitrogen species discovered under high-pressure (e.g. pentazolate-anion, whose first stabilization in bulk was achieved in $CsN_5$ at 60 GPa[22]) were subsequently synthesized by conventional chemistry methods under ambient pressure[34–36].

In addition to the discoveries of unique nitrogen entities that push the boundaries of fundamental nitrogen chemistry, nitrides and polynitrides synthesized under high pressure often possess key properties for functional applications such as high hardness[7], unique electronic properties[33], and high energy density[37]. Polynitrides with a high nitrogen content are especially promising as high-energy-density materials (HEDM) because their decomposition results in the

[1]Bavarian Research Institute of Experimental Geochemistry and Geophysics (BGI), University of Bayreuth, 95440 Bayreuth, Germany. [2]Material Physics and Technology at Extreme Conditions, Laboratory of Crystallography, University of Bayreuth, 95440 Bayreuth, Germany. [3]Centre for Science at Extreme Conditions and School of Physics and Astronomy, University of Edinburgh, EH9 3FD Edinburgh, United Kingdom. [4]Department of Physics, Chemistry and Biology (IFM), Linköping University, SE-581 83 Linköping, Sweden. [5]Center for Advanced Radiation Sources, University of Chicago, Chicago, IL 60637, USA. [6]European Synchrotron Radiation Facility, 38000 Grenoble, France. ✉e-mail: andrii.aslandukov@uni-bayreuth.de

formation of molecular nitrogen, which is accompanied by a large energy release. The latter is due to a large difference between the energy of the triple intramolecular bond in $N_2$ and the energy of double and single bonds in polynitrogen units[37]. For HEDMs the molecular weight of the compound also matters: with other properties being similar, the lighter the elements in the solid, the higher the gravimetric energy density of the compound. Since scandium is the lightest transition metal, its polynitrides may be especially promising as HEDM.

Hitherto, only one binary Sc-N compound is known: cubic scandium nitride ScN with the rock salt structure, which exists at ambient conditions and is predicted to be stable up to ~250 GPa[38]. There are several theoretical studies[39–42], where nitrogen-rich phases with $ScN_3$, $ScN_5$, $ScN_6$, and $ScN_7$ compositions have been predicted to be stable under 30–110 GPa and may have potential as HEDM (gravimetric energy density ranges from 2.40 kJ/g to 4.23 kJ/g).

In this study, we experimentally investigated the behavior of the Sc-N system at pressures between 50 to 125 GPa and high temperatures. Here we present the synthesis and characterization of four novel Sc-N phases, whose structures were solved and refined on the basis of single-crystal X-ray diffraction. The nitrogen-rich polynitrides $Sc_2N_6$, $Sc_2N_8$, and $ScN_5$ feature a unique nitrogen catenation: previously unknown $N_6^{6-}$ and $N_8^{6-}$ nitrogen units and $_\infty^2(N_5^{3-})$ anionic 2D-polynitrogen layers consisting of fused $N_{12}$ rings, respectively.

## Results and discussion

In this study diamond anvil cells (DACs) loaded with scandium pieces embedded in molecular nitrogen were used (see Methods section for details). Samples were compressed to their target pressures and laser-heated at 2500(300) K. Laser-heating experiments were carried out at pressures of 50(1), 78(2), 96(2), and 125(2) GPa (Supplementary Table 1). After laser-heating, detailed X-ray diffraction maps were collected around the heated spot to pinpoint the location of crystallites most appropriate for single-crystal X-ray diffraction measurements (Fig. 1). Then single-crystal X-ray diffraction data (Supplementary Fig. 1) were collected at the selected positions to identify the phases' crystal structure and chemical composition.

According to the synchrotron single-crystal X-ray diffraction data, only the well-known ScN phase (rock-salt type structure, $a$ = 4.2492(7) Å, V = 76.72(4) Å³ at 50 GPa) was formed at 50 GPa. The obtained volume is in good agreement with the published ScN equation of state[38]. At 78 GPa, two novel scandium nitrides with chemical formulas $Sc_2N_6$ and $Sc_2N_8$ were obtained along with ScN. At 96 GPa, a mixture of ScN, $Sc_2N_8$, as well as the previously unobserved $ScN_5$, was obtained. And, finally, at 125 GPa the collected synchrotron

single-crystal X-ray diffraction data and the subsequent crystal structure solution and refinement revealed the formation of the $ScN_5$ and $Sc_4N_3$ phases. Overall four novel Sc-N phases were synthesized by chemical reactions of Sc and $N_2$ at 2500 K in the pressure range of 78 to 125 GPa (Supplementary Fig. 2).

Remarkably, at 50 GPa, scandium behaves like at ambient pressure producing only ScN, while at higher pressures a rich variety of phases was observed. In addition to a significant increase in the chemical potential of nitrogen under high pressure[43], another possible reason explaining such difference in chemistry between 50 and 78 GPa is a significant drop in scandium's electronegativity at 60 GPa (Supplementary Fig. 3a) and as a result, scandium is predicted to be the least electronegative atom in 60–110 GPa pressure range[44]. It leads to the significant increase of difference in electronegativity between N and Sc above 60 GPa (Supplementary Fig. 3b), which increases the chemical reactivity of scandium, decreases the potential kinetic barriers of reactions, and leads to the appearance of more local minima in the energy landscape.

The refinement against single-crystal X-ray diffraction data for all synthesized compounds resulted in very good agreement factors (Supplementary Tables 2–6). For cross-validation of the crystal structures, we performed density functional theory (DFT) calculations using the Vienna ab initio simulation package[45] (see Methods section for details). We carried out variable cell structural relaxations for $Sc_2N_6$, $Sc_2N_8$, and $ScN_5$ and found that the relaxed structural parameters closely reproduce the corresponding experimental values (Supplementary Tables 7–9).

$Sc_2N_6$ synthesized at 78 GPa (Fig. 2a) crystalizes in the triclinic crystal system (space group $P\bar{1}$ (#2)). The structure of $Sc_2N_6$ has one Sc and three N distinct atomic positions (see Supplementary Table 3 and the CIF for the full crystallographic data). Nitrogen atoms form isolated "zig-zag" $N_6$ units (Fig. 2a, b). The existence of this phase was predicted at pressures of 30–100 GPa[39].

The structure of $Sc_2N_8$ (Fig. 2d) has the monoclinic space group $P2_1/c$ (#14) with one Sc and four N distinct atomic positions (see Supplementary Table 4 and the CIF for the full crystallographic data). Nitrogen atoms form isolated "zig-zag" $N_8$ units (Fig. 2d, e) that have never been observed or predicted.

The bond length analysis of the $N_6$ and $N_8$ units suggests that N1-N2, N2-N3 (in $N_6$ unit) and N1-N2, N2-N3, N4-N4 (in $N_8$ unit) are single-bonded, while N3-N4 (in $N_6$ unit) and N3-N4 (in $N_8$ unit) are double-bonded (Fig. 2b,c,e,f). Then, the charges can be described in a classic ionic approach: the total charge of $[N_6]^{6-}$ and $[N_8]^{6-}$ units is 6-, which corresponds to the +3 oxidation state of Sc atoms. The angle values and a small difference in bond length indicate the strong electron delocalization (indeed several different resonance Lewis formulas can

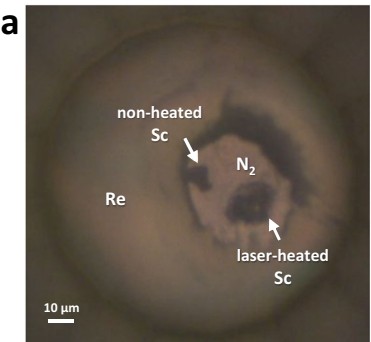

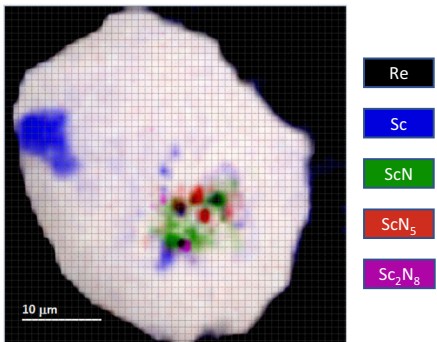

**Fig. 1 | Sample chamber of the diamond anvil cell at 96 GPa. a** Micro-photo of the sample chamber. **b** 2D X-ray diffraction map (collected with 0.75 μm steps at the ID11 beamline of the ESRF) showing the distribution of the scandium nitrides phases (determined by single-crystal XRD) within the heated sample at 96 GPa.

The color intensity is proportional to the intensity of the following reflections: the (1 1 1) and (3 3 1) of ScN for the green regions; the (1 1 1) of $ScN_5$ for the red regions; the (0 2 1) of $Sc_2N_8$ for the purple regions.

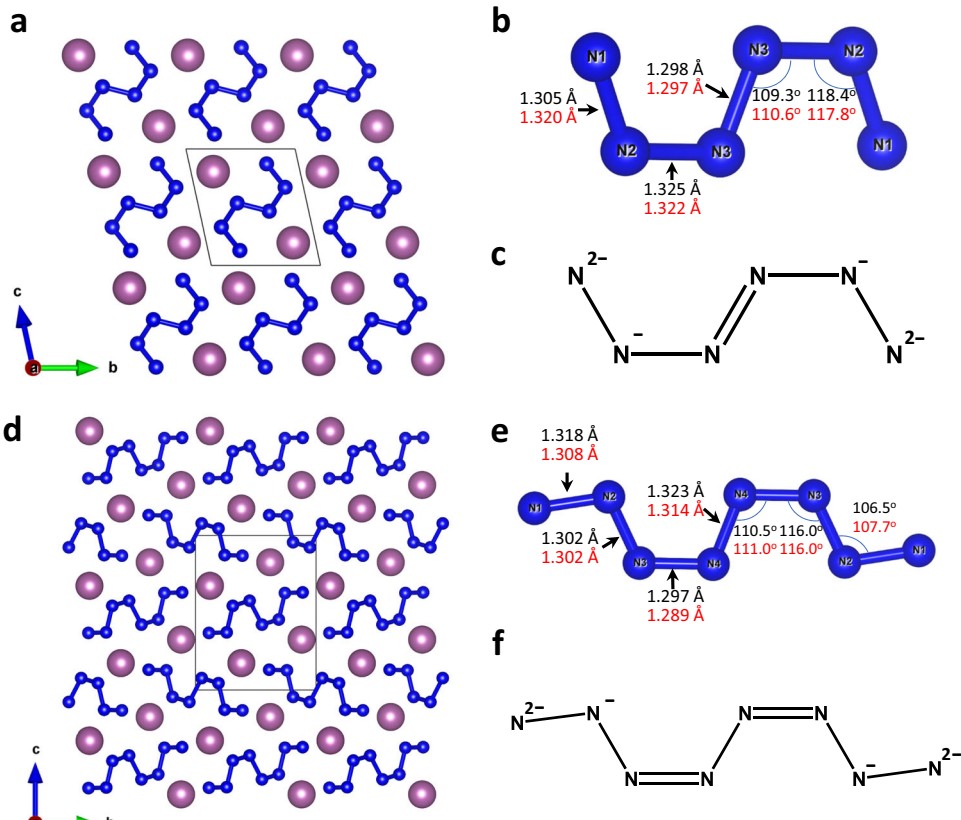

**Fig. 2 | Crystal structure of $Sc_2N_6$ and $Sc_2N_8$ at 78 GPa. a** A view of $Sc_2N_6$ along the *a*-axis; **b** an $N_6$ unit; **c** structural formula of an $N_6$ unit; **d** a view of $Sc_2N_8$ along the *a*-axis; **e** an $N_8$ unit; **f** structural formula of an $N_8$ unit. Sc atoms are purple, N atoms are blue; thin grey lines outline the unit cell. Values of bond lengths and angles obtained from the experiment are shown in black, while those obtained from the DFT calculations are shown in red.

be drawn for $N_6$ and $N_8$) and nitrogen atoms cannot be considered as purely $sp^2$ or $sp^3$ hybridized.

The two novel catenated nitrogen units $N_6^{6-}$ and $N_8^{6-}$ discovered in this study—being intermediate non-cyclic species between dinitride and 1D-polynitrogen anions—significantly expand the list of anionic nitrogen oligomers (Fig. 3). Notably, all these units are built of an even number of nitrogen atoms suggesting their formation via the polymerization of dinitrogen molecules. The degree of polymerization increases with pressure: dinitrides are synthesized at low pressures (<50 GPa); $N_4$, $N_6$, $N_8$ units are obtained at mild pressures (50–80 GPa), while 1D-polynitrogen chains are usually formed above 100 GPa. Since the dinitrogen ($[N_2]^{x-}$ x = 0.66, 0.75, 1, 2, 3, 4), and 1D-polynitrogen ($[N_4]_\infty^{x-}$, $x = 2$–6) anions are able to accumulate different charges, one can expect that the $N_6$ and $N_8$ units can also exist in different charge states, and therefore can be found in other metal-nitrogen systems.

The structure of $ScN_5$ has the monoclinic space group $P2_1/m$ (#11) with one Sc and three N distinct atomic positions (see Supplementary Table 5 and the CIF for the full crystallographic data). Nitrogen atoms form corrugated 2D polymeric $_\infty^2(N_5^{3-})$ layers alternating along the *a*-axis built of fused $N_{12}$ rings (Fig. 4a). Sc atoms are located in between the layers, in the way that the projection of Sc atoms along the *a*-axis is in the center of the $N_{12}$ rings (Fig. 4b). Sc atoms are eight-fold coordinated (coordination number CN = 8, coordination polyhedron is a distorted square antiprism) by four N atoms of the lower layer and four N atoms of the upper layer (Fig. 4c).

The analysis of N-N lengths in $ScN_5$ suggests that all N-N bonds are single bonds (Fig. 4d). All N atoms can be considered as $sp^3$-hybridized, which also explains that the values of N-N-N angles in the $N_{12}$ cycles are close to the ideal tetrahedra angle (98.7°–114.5°, Fig. 4e). N1 atoms make three covalent N-N bonds, while N2 and N3 atoms make only two,

therefore one can suggest a −1 charge on the N2 and N3 atoms. It corresponds to the +3 oxidation state of Sc atoms.

Despite the theoretical prediction of four different structures with the $ScN_5$ composition[39–41], the here observed structure was not predicted. Usually in polynitrides nitrogen prefers to form 1D polymeric chains[20,28–33], and among all the experimentally synthesized polynitrides up-to-date there is only one discovered polynitride with 2D polynitrogen layers—monoclinic $BeN_4$[33] with layers consisting of the fused $N_{10}$ rings. The polynitrogen layers in $ScN_5$ can be considered as distorted bp-N layers[2], where 1/6 atoms are missing (Supplementary Fig. 4).

$ScN_5$ is isostructural to a family of polyphosphides $LnP_5$ (Ln=La-Lu, Y (except Eu and Pm)) known at ambient conditions[46,47]. It fully obeys the ninth high-pressure chemistry rule of thumb formulated in 1998: *"Elements behave at high pressures like the elements below them in the periodic table at lower pressures"*[48]. The adoption of this structure type is also advantageous from a geometric point of view, since the ratio of ionic radii r($N^{3-}$)/r($Sc^{3+}$)=1.97 in $ScN_5$ perfectly fits r($P^{3-}$)/r($Y^{3+}$) = 1.95 in the above-mentioned family member $YP_5$.

$Sc_4N_3$ synthesized at 125 GPa has a well-known anti-$Th_3P_4$ structure type (space group $I\bar{4}3d$ (#220)) and contains only distinct, not-catenated N atoms (see Supplementary Table 6, Supplementary Fig. 5, and the CIF for the full crystallographic data), which we do not discuss in detail here. This $Sc_4N_3$ structure was predicted to be thermodynamically stable above 80 GPa[39].

In order to get a deeper insight into the chemistry and the physical properties of the novel compounds, further DFT calculations were performed (see Methods section for details). As mentioned above, variable-cell structural relaxations for the $Sc_2N_6$, $Sc_2N_8$, and $ScN_5$ compounds at the synthesis pressure closely reproduced structural

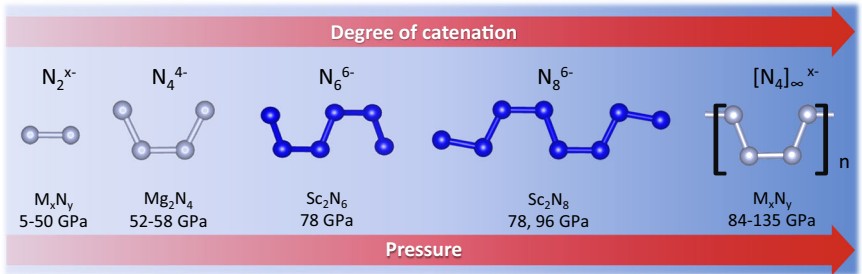

**Fig. 3 | Experimentally observed catenated nitrogen units and chains.** The units in blue were first discovered in the present study.

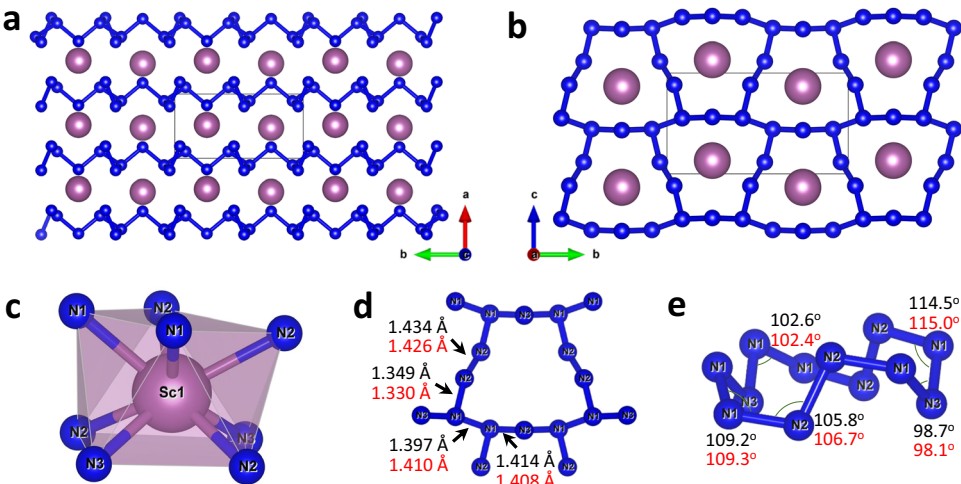

**Fig. 4 | Crystal structure of ScN₅ at 96 GPa. a** A view of the crystal structure along the *c*-axis. **b** A view of the crystal structure along the *a*-axis. **c** The coordination environment of the Sc atom. **d** A specific view of N₁₂ cycle along the *a*-axis. **e** A general view of N₁₂ cycle. Sc atoms are purple, N atoms are blue; thin grey lines outline the unit cell. Values of bond lengths and angles obtained from the experiment are shown in black, while those obtained from the DFT calculations are shown in red.

parameters and bond lengths obtained from the experimental data. The phonon dispersion relations calculated in the harmonic approximation show that $Sc_2N_6$, $Sc_2N_8$, and $ScN_5$ phases are dynamically stable at 96 GPa and remain dynamically stable at ambient pressure (Supplementary Figs. 6–8). Considering dynamical stability at 1 bar, we have attempted to quench $Sc_2N_6$, $Sc_2N_8$, $ScN_5$ phases, however, due to technical limitations of the decompression experiment (see footnote Supplementary Table 1), no conclusion regarding their recoverability could be made. To trace the structures' behavior during the pressure release and to get the equations of state of all synthesized nitrogen-rich high-pressure scandium polynitrides, the full variable-cell structure relaxation for the $Sc_2N_6$, $Sc_2N_8$, and $ScN_5$ compounds were performed with 10 GPa pressure steps in the range of 0–150 GPa (Supplementary Fig. 9). The volume-pressure dependences of DFT-relaxed structures of $Sc_2N_6$, $Sc_2N_8$, and $ScN_5$ were fitted with a 3$^{rd}$ order Birch-Murnaghan equation of state (Supplementary Fig. 10). The obtained bulk moduli $(K_0(Sc_2N_6) = 160$ GPa, $K_0(Sc_2N_8) = 173$ GPa, $K_0(ScN_5) = 205$ GPa) are lower than or comparable to the bulk modulus of known ScN $(K_0(ScN) = 207$ GPa)[38].

Under the same pressure, the volume per atom for all investigated nitrides monotonously linearly decreases with increasing nitrogen content (Supplementary Fig. 11a). Interestingly, the volume per nitrogen atom in the $ScN$-$Sc_2N_6$-$Sc_2N_8$-$ScN_5$ series does not decrease with the degree of nitrogen polymerization (Supplementary Fig. 11b), so nitrogen polymerization probably is a way of crystal structure adaptation to closer N-N contacts.

While the structure of $Sc_2N_6$ has been predicted[39], the crystal structures of $Sc_2N_8$ and $ScN_5$ we observed have not been predicted.

Remarkably, four different crystal structures with the $ScN_5$ composition were proposed[39–41], but the one we synthesized in the present study ($P2_1/m$ $ScN_5$) was not among them. Our calculations of relative formation enthalpies of $ScN_5$ for various predicted structures ($Cm$ $ScN_5$[39], $P{-}1$ $ScN_5$[39], $C2/m$ $ScN_5$,[40] and $P2_1/c$ $ScN_5$[41]) with respect to $P2_1/m$ $ScN_5$ (Supplementary Fig. 12a) in the range of 0 to150 GPa have shown that above 46 GPa the $P2_1/m$ $ScN_5$ phase is thermodynamically more stable than all other predicted phases. Below 46 GPa $P{-}1$ $ScN_5$[39] is more favorable. The $C2/m$ $ScN_5$[40] and $P2_1/c$ $ScN_5$[41] phases are not energetically competitive with $P2_1/m$ $ScN_5$ in the whole pressure range studied (Supplementary Fig. 12a).

To estimate the thermodynamic stability of the $Sc_2N_6$, $Sc_2N_8$, and $ScN_5$ phases, the nitrogen-rich part of the static enthalpy convex hull was calculated at different pressures. $Sc_2N_6$ and $ScN_5$ phases were found to be stable at the synthesis pressures (78 and 96 GPa, Supplementary Fig. 13a and Supplementary Fig. 12b), but $Sc_2N_8$ appears to be out of the convex hull (40 meV and 50 meV per atom above the convex hull at 78 and 96 GPa, respectively). Such insignificant departures from the convex hull, smaller than $k_BT$ at the synthesis temperature (2500 K, 215 meV), suggest that $Sc_2N_8$ may be thermodynamically stable at high temperatures and preserved as a metastable state under rapid T-quench to room temperature. $ScN_5$ remains thermodynamically stable at least down to 40 GPa (Supplementary Fig. 13b), and $Sc_2N_6$− down to 30 GPa (Supplementary Fig. 13c), while at 20 GPa all nitrogen-rich scandium phases are out of the convex-hull (Supplementary Fig. 13d).

The calculated electron localization functions for $Sc_2N_6$, $Sc_2N_8$, and $ScN_5$ demonstrate a strong covalent bonding between nitrogen

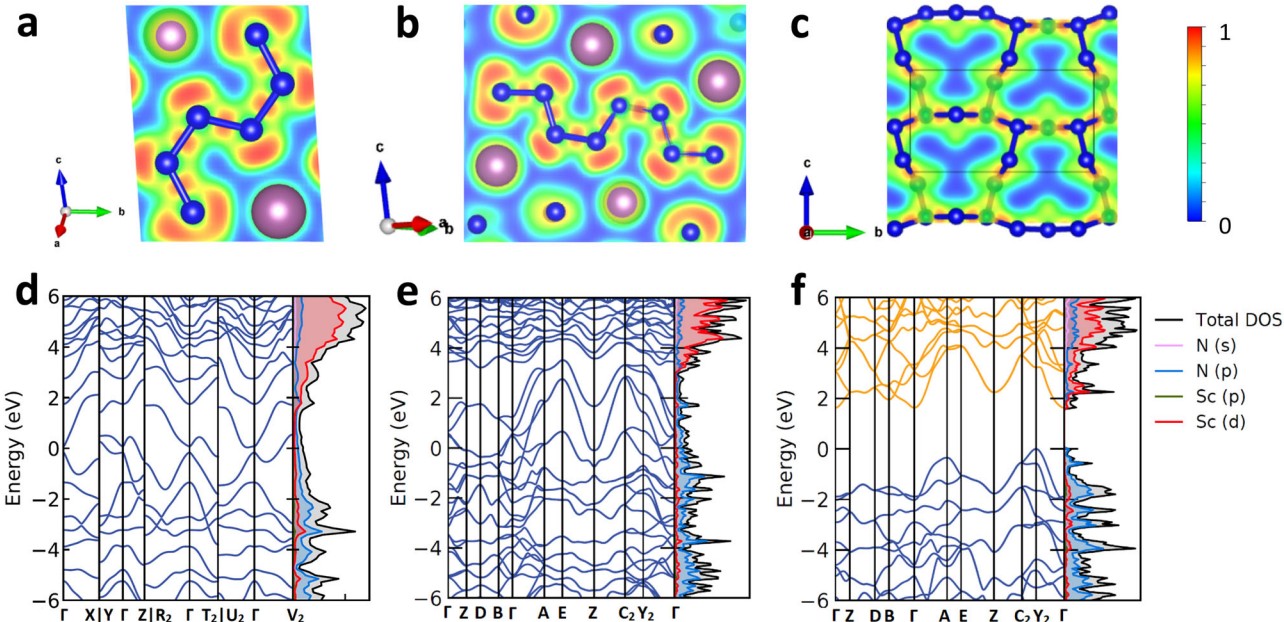

**Fig. 5 | Calculated electronic properties of $Sc_2N_6$ at 78 GPa, and $Sc_2N_8$, $ScN_5$ at 96 GPa.** Electron localization function calculated for (**a**) $Sc_2N_6$ in the (3 0 2) plane, (**b**) $Sc_2N_8$ in the (−2 4 1) plane, and (**c**) $ScN_5$ in the (1 0 0) plane. The electron density of states of (**d**) $Sc_2N_6$, (**e**) $Sc_2N_8$, and (**f**) $ScN_5$.

atoms within the $N_6$, $N_8$ units, and 2D-polynitrogen layers (Fig. 5a–c), while there is no covalent bonding between nitrogen and scandium atoms. The computed electron density of states (DOS) shows that $Sc_2N_6$ and $Sc_2N_8$ are metals (Fig. 5d, e) with an anion-driven metallicity[10], since the main electronic contribution at the Fermi level comes from the nitrogen $p$-states. At the same time, $ScN_5$ is an indirect semiconductor with a band gap of 1.8 eV at 96 GPa (Fig. 5f). One can explain such different electronic properties considering the chemical bonding in these compounds. In $ScN_5$ there are only single N-N bonds, which means all π* antibonding nitrogen molecular orbitals are fully filled, whereas, in $Sc_2N_6$ and $Sc_2N_8$, containing delocalized π-bonds within $N_6^{6-}$ and $N_8^{6-}$ units, π* antibonding nitrogen states are partially filled and can conduct electrons through the π*-band. A similar trend of electronic properties with respect to the presence of N-N π-bonds is observed for many known polynitrides[27–33]. Among all known polynitrides there are only two compounds with solely σ N-N bonds: $TaN_5$, which contains single-bonded branched polynitrogen chains[31], and m-$BeN_4$, which contains single-bonded 2D-polynitrogen layers[33]. Both compounds are semiconductors, as reported for $TaN_5$[31], and calculated for m-$BeN_4$ in the present study (Supplementary Fig. 14). Other polynitrides contain N-N π-bonds and the majority of them (tr-$BeN_4$, $FeN_4$, α-$ZnN_4$, β-$ZnN_4$, $TaN_4$, $ReN_8$·x$N_2$, $WN_8$·$N_2$, $Os_5N_{28}$·3$N_2$, $Hf_4N_{20}$·$N_2$, $Hf_2N_{11}$, $Y_2N_{11}$, $YN_6$)[27–33] exhibit an anion-driven metallicity.

Considering the dynamical stability of $Sc_2N_6$, $Sc_2N_8$, and $ScN_5$ at ambient pressure, these phases might be preserved at ambient conditions as metastable and potentially can serve as high-energy-density materials. The key metrics of energetic materials' performance[49], such as volumetric and gravimetric energy densities, detonation velocity, and detonation pressure, were estimated for $Sc_2N_6$, $Sc_2N_8$, and $ScN_5$ (Table 1) considering their decomposition to ScN and molecular nitrogen at 1 bar (see Methods section for details).

The energy densities and explosive performance increase from $Sc_2N_6$ to $ScN_5$ along with the increase in nitrogen content. Due to the higher density of scandium nitrides compared to organic explosives, they possess extremely high volumetric energy densities that are higher than the typical energy density of TNT. The estimated gravimetric energy densities are lower than that of TNT, but higher than those of many other polynitrides[31] since scandium is a light metal. The estimated detonation velocity and detonation pressure of scandium

polynitrides are also higher than those of TNT. Thus, the $Sc_2N_6$, $Sc_2N_8$, and $ScN_5$ are promising high-energy-density materials.

To summarize, in this study, four novel Sc-N phases—$Sc_2N_6$, $Sc_2N_8$, $ScN_5$, and $Sc_4N_3$—were synthesized from Sc and $N_2$ by laser-heating at 2500 K at pressures between 78 and 125 GPa. Nitrogen-rich scandium polynitrides $Sc_2N_6$, $Sc_2N_8$, and $ScN_5$ demonstrate a unique nitrogen catenation: they feature $N_6$ units, $N_8$ units, and 2D polynitrogen $^2_\infty(N_5^{3-})$ layers consisting of $N_{12}$ fused rings, respectively. DFT calculations showed that all three scandium polynitrides are dynamically stable at the synthesis pressure as well as at 1 bar. $Sc_2N_6$ and $Sc_2N_8$ are metals with the main electronic contribution at the Fermi level that comes from the nitrogen $p$-states, while $ScN_5$ is an indirect semiconductor. Synthesized $Sc_2N_6$, $Sc_2N_8$, and $ScN_5$ compounds are promising high-energy-density materials with volumetric energy densities, detonation velocities, and detonation pressures higher than those of TNT.

One can expect that the $N_6$ and $N_8$ units will be stabilized at ambient conditions in the future, considering a positive example of $CsN_5$ high-pressure synthesis and subsequent stabilization of the $N_5^-$ anion at atmospheric pressure. It may not only open access to novel high-energy-density materials but also to analogues of Li- and Mg-metalorganic compounds that are currently widely used in organic synthesis. $N_6$ and $N_8$ units, if used as building blocks in organic chemistry, may provide new routes for the targeted synthesis of novel N-heteroatomic organic, metalorganic, and coordination compounds.

## Methods

### Sample preparation

The BX90-type large X-ray aperture DACs[52] equipped with Boehler-Almax type diamonds[53] (culet diameters are 250, 120, and 80 μm) were used in the experiments. The sample chambers were formed by pre-indenting of rhenium gaskets to 20, 18, and 15 μm thickness and laser-drilling a hole of 115, 60 and 40 μm, respectively, in diameter in the center of the indentation. A DAC equipped with 250-μm culet anvils was used for the experiment at 50(1) GPa; a DAC equipped with 120-μm culet anvils was used for experiments at 78(2) and 96(2); and a DAC equipped with 80-μm culet anvils was used for the experiment at 125(2) GPa. A piece of scandium (99.9%, Sigma Aldrich) was placed in a sample chamber, then molecular nitrogen (purity grade N5.0) was

**Table 1 | Characteristics of $Sc_2N_6$, $Sc_2N_8$, $ScN_5$ and TNT as energetic materials**

| Compound | Density $\rho$, g/cm³ | Energy density | | Detonation velocity $V_d$, km/s | Detonation pressure $P_d$, GPa |
|---|---|---|---|---|---|
| | | gravimetric GED, kJ/g | volumetric VED, kJ/cm³ | | |
| $Sc_2N_6$ | 3.65 | 2.28 | 8.31 | 6.9 | 30 |
| $Sc_2N_8$ | 3.58 | 3.07 | 11.0 | 8.3 | 43 |
| $ScN_5$ | 3.71 | 3.76 | 14.0 | 9.8 | 60 |
| TNT | 1.64[50] | 4.3[51] | 7.2[51] | 6.9[50] | 19[50] |

loaded using a BGI high-pressure gas loading system (1300 bars)[54]. The sizes of the scandium pieces were $40 \times 40 \times 8$ μm³ for 250 μm culet anvils and not bigger than $15 \times 15 \times 5$ μm³ for DACs with anvils of all other sizes. The samples were compressed to target pressure (50(1), 78(2), 96(2), and 125(2) GPa) and then laser-heated up to 2500(200) K using a home-made double-sided laser-heating system equipped with two YAG lasers ($\lambda = 1064$ nm) and the IsoPlane SCT 320 spectrometer with a $1024 \times 2560$ PI-MAX 4 camera for the collection of thermal emission spectra from the heated spot[55]. The temperature during the laser heating was determined by fitting of sample's thermal emission spectra to the grey body approximation of Planck's radiation function in a given wavelength range (570–830 nm). The pressure in the DACs was determined using the Raman signal from the diamond anvils[56] and monitored additionally by X-ray diffraction of the Re gasket edge using the rhenium equation of state[57].

## X-ray diffraction

The X-ray diffraction studies were done at the ID11 beamline ($\lambda = 0.2843$ Å and $\lambda = 0.2846$ Å) and ID15b beamline ($\lambda = 0.4100$ Å) of the Extreme Brilliant Source European Synchrotron Radiation Facility (EBS-ESRF) as well as at the GSECARS 13IDD beamline of the APS ($\lambda = 0.2952$ Å). At ID11 beamline of ESRF the X-ray beam was focused down to $0.75 \times 0.75$ μm² and data was collected with Eiger2X CdTe 4 M hybrid photon counting pixel detector. At ID15b beamline of ESRF the X-ray beam was focused down to $1.5 \times 1.5$ μm² and data was collected with Eiger2X CdTe 9 M hybrid photon counting pixel detector. At 13IDD beamline of APS the X-ray beam was focused down to $2 \times 2$ μm² and data was collected with Pilatus 1 M detector. In order to determine the position of the polycrystalline sample on which the single-crystal X-ray diffraction acquisition is obtained, a full X-ray diffraction mapping of the pressure chamber was achieved. The sample position displaying the most and the strongest single-crystal reflections belonging to the phase of interest was chosen for the collection of single-crystal data, collected in step-scans of 0.5° from −36° to +36°. The CrysAlis$^{Pro}$ software package[58] was used for the analysis of the single-crystal XRD data (peak hunting, indexing, data integration, frame scaling, and absorption correction). To calibrate an instrumental model in the CrysAlis$^{Pro}$ software, i.e., the sample-to-detector distance, detector's origin, offsets of the goniometer angles, and rotation of both the X-ray beam and detector around the instrument axis, we used a single crystal of orthoenstatite [$(Mg_{1.93}Fe_{0.06})(Si_{1.93},Al_{0.06})O_6$, $Pbca$ space group, $a = 8.8117(2)$ Å, $b = 5.18320(10)$ Å, and $c = 18.2391(3)$ Å]. The DAFi program was used for the search of reflection's groups belonging to the individual single crystal domains[59]. Using the OLEX2 software package[60], the structures were solved with the ShelXT structure solution program[61] using intrinsic phasing and refined with the ShelXL[62] refinement package using least-squares minimization. Crystal structure visualization was made with the VESTA software[63].

## Theoretical calculations

First-principles calculations were performed using the framework of density functional theory (DFT) as implemented in the Vienna Ab initio Simulation Package (VASP)[64]. The Projector-Augmented-Wave (PAW) method[65] was used to expand the electronic wave function in plane waves. The Generalized Gradient Approximation (GGA) functional is used for calculating the exchange-correlation energies, as proposed by Perdew−Burke−Ernzerhof (PBE)[66]. The recommended PAW potentials "Sc_sv" and "N" with the following valence configurations of $3s^23p^64s^23d^1$ for Sc and $2s^22p^3$ for N were used. We used the Monkhorst−Pack scheme with $10 \times 10 \times 10$ for ScN, $12 \times 8 \times 8$ for $Sc_2N_6$, $10 \times 6 \times 4$ for $Sc_2N_8$, $12 \times 6 \times 12$ for $ScN_5$ $k$-points for Brillouin zone sampling, and the plane-wave kinetic energy cutoff was set to 800 eV, with which total energies are converged to better than 2 meV/atom. The electronic convergence criterion was set to $\Delta E = 10^{-8}$ eV, this minimized the interatomic forces to $F_{atom} < 10^{-3}$ eV/Å. For electron band structure calculations the 1.5−2 fold denser $k$-points grids were used. The finite displacement method, as implemented in PHONOPY[67], was used to calculate phonon frequencies and phonon band structures. The 4×3×3, 3×2×2, and $3 \times 2 \times 3$ supercells with $4 \times 4 \times 4$ $k$-points grids for $Sc_2N_6$, $Sc_2N_8$, and $ScN_5$, respectively, were used for phonon calculations and displacement amplitudes were of 0.01 Å.

The gravimetric and volumetric energy densities of $Sc_2N_6$, $Sc_2N_8$, and $ScN_5$ were calculated considering the enthalpy change ΔH for the following chemical decomposition reactions at ambient pressure at 0 K (the $Fm$−$3m$-ScN and $\alpha$-$N_2$ structures of products were used in the calculations since they are the most stable polymorphs at such conditions):

$$Sc_2N_6 \rightarrow 2\,ScN + 2\,N_2$$

$$Sc_2N_8 \rightarrow 2\,ScN + 3\,N_2$$

$$ScN_5 \rightarrow ScN + 2\,N_2$$

The detonation velocity ($V_d$, km/s) and detonation pressure ($P_d$, GPa) of the $Sc_2N_6$, $Sc_2N_8$, and $ScN_5$ were estimated by the Kamlet-Jacobs empirical equations[50]:

$$V_d = (N \cdot M^{0.5} \cdot GED^{0.5})^{0.5} \cdot (1.011 + 1.312\rho) \tag{1}$$

$$P_d = 1.588 \cdot N \cdot M^{0.5} \cdot GED^{0.5} \cdot \rho^2 \tag{2}$$

where $N$ is the number of moles of gaseous detonation product (nitrogen gas) per gram of explosive, $M$ is the molar mass (28 g/mol) of nitrogen gas, $GED$ is gravimetric energy density in cal/g, and $\rho$ is density in g/cm³.

## Data availability

The full crystallographic data for structures reported in this article have been deposited at the Inorganic Crystal Structure Database (ICSD) under deposition numbers CSD 2252030−2252036. These data can be obtained from CCDC's and FIZ Karlsruhe's free service for viewing and retrieving structures (https://www.ccdc.cam.ac.uk/structures/). The crystallographic information (CIF files, FCF files, and the corresponding CheckCIF reports) is also available as Source data. All other datasets generated and/or analyzed during the current

study are available from the corresponding author upon request. Source data are provided with this paper.

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

## Acknowledgements

The authors thank Prof. Björn Winkler for useful discussions. The authors acknowledge the Advanced Photon Source (APS) for the provision of beamtime at the 13ID-D beamline and the European Synchrotron Radiation Facility (ESRF) for the provision of beamtime at the ID11 and ID15b beamlines. Portions of this work were performed at GeoSoilEnviroCARS (The University of Chicago, Sector 13), Advanced Photon Source (APS), Argonne National Laboratory. GeoSoilEnviroCARS was supported by the National Science Foundation – Earth Sciences (EAR – 1634415). This research used resources of the Advanced Photon Source, a U.S. Department of Energy (DOE) Office of Science User Facility operated for the DOE Office of Science by Argonne National Laboratory under Contract No. DE-AC02-06CH11357. Computations were performed at the Leibniz Supercomputing Center of the Bavarian Academy of Sciences and the Humanities, and the research center for scientific computing at the University of Bayreuth. D.L. thanks the UKRI Future Leaders Fellowship (MR/V025724/1) for financial support. N.D. and L.D. thank the Deutsche Forschungsgemeinschaft (DFG projects DU 945/15-1, LA 4916/1-1, DU 954–11/1, DU 393–9/2, DU 393–13/1) for financial support. N.D. also thanks the Swedish Government Strategic Research Area in Materials Science on Functional Materials at Linköping University (Faculty Grant SFO-Mat-LiU No. 2009 00971). For the purpose of open access, the authors have applied a Creative Commons Attribution (CC BY) licence to any Author Accepted Manuscript version arising from this submission. Open access is funded by the Open Access Publishing Fund of the University of Bayreuth.

## Author contributions

An.A., L.D., and N.D. designed the research. An.A. and Al.A. prepared the high-pressure experiments. An.A., Al.A., D.L., S.K., Y.Y., F.I.A., S.C., V.P., E.L.B., C.G., J. W., D.C., M.H. performed the synchrotron X-ray diffraction experiments. An.A. processed the synchrotron X-ray diffraction data. An.A. and Al.A. performed the theoretical calculations. An.A. and L.D. contextualized the data interpretation. An.A., L.D., and N.D. prepared the first draft of the paper with contributions from all other authors. All the authors commented on successive drafts and have given approval to the final version of the paper.

## Funding

## Competing interests

The authors declare no competing interests.
