## [Peer Review File · Nature Communications]

REVIEWERS' COMMENTS

Reviewer #1 (Remarks to the Author):

This study reports the high-pressure, high-temperature synthesis and characterization of four new scandium nitride compounds - Sc₂N₆, Sc₂N₈, ScN₅, and Sc₄N₃. Single-crystal X-ray diffraction reveals that the nitrogen-rich Sc₂N₆, Sc₂N₈, and ScN₅ phases contain unique nitrogen catenation, with previously unknown N₆ and N₈ units, and corrugated 2D polymeric layers of fused N₁₂ rings, respectively. Density functional theory calculations confirm their dynamical stability and metallicity for Sc₂N₆/Sc₂N₈, versus semiconducting behavior for ScN₅. Estimations highlight their promise as high-energy density materials, with volumetric energy densities, detonation velocities, and pressures exceeding TNT, aided by scandium's light atomic weight. Overall, these scandium nitride phases significantly expand nitrogen chemistry and the family of polynitrogen anions, while demonstrating advanced functional properties relevant for energetic materials. Accordingly, I recommend its publication without further revisions.

Reviewer #2 (Remarks to the Author):

I appreciate the work that the authors have put into improving the manuscript, and recommend publication in Nature Communications with no further changes. I have just two comments:

(1) The authors write:

On page 10260 one reads: "A higher reactivity and polarity would be anticipated in cases where the difference in electronegativity between constituent atoms increases upon compression."

Indeed, and a discussion of the difference in electronegativity between Sc and N at high pressures would be insightful and a valuable addition to the manuscript. My comment was that it is not sufficient to simply state that the decrease in electronegativity of Sc is what leads to its increased reactivity, without comparing it directly to that of nitrogen. One could do this quantitatively using the reference at hand.

(2) The authors write:

We provided CIF files for all new phases discussed in the paper upon the initial submission, and the quality of experimental structural data can be easily accessed through the CIF files.

Indeed, and my apologies for missing this the first time around. My age is catching up to me. I agree with the authors that these data are of high quality, and stand by my suggestion to include representative images of the raw data, which show extraordinary diffraction quality for DAC work.

Reply to the Reviewers' comments

Reviewer #1 (Comments for the Author):

This study reports the high-pressure, high-temperature synthesis and characterization of four new scandium nitride compounds - Sc₂N₆, Sc₂N₈, ScN₅, and Sc₄N₃. Single-crystal X-ray diffraction reveals that the nitrogen-rich Sc₂N₆, Sc₂N₈, and ScN₅ phases contain unique nitrogen catenation, with previously unknown N₆ and N₈ units, and corrugated 2D polymeric layers of fused N₁₂ rings, respectively. Density functional theory calculations confirm their dynamical stability and metallicity for Sc₂N₆/Sc₂N₈, versus semiconducting behavior for ScN₅. Estimations highlight their promise as high-energy density materials, with volumetric energy densities, detonation velocities, and pressures exceeding TNT, aided by scandium's light atomic weight. Overall, these scandium nitride phases significantly expand nitrogen chemistry and the family of polynitrogen anions, while demonstrating advanced functional properties relevant for energetic materials. Accordingly, I recommend its publication without further revisions.

Author reply: We thank Reviewer #1 for her/his positive evaluation of our work.

Reviewer #2 (Comments for the Author):

I appreciate the work that the authors have put into improving the manuscript, and recommend publication in Nature Communications with no further changes. I have just two comments:

Author reply: We thank Reviewer #2 for her/his positive evaluation of our work.

(1) The authors write: On page 10260 one reads: "A higher reactivity and polarity would be anticipated in cases where the difference in electronegativity between constituent atoms increases upon compression."

Indeed, and a discussion of the difference in electronegativity between Sc and N at high pressures would be insightful and a valuable addition to the manuscript. My comment was that it is not sufficient to simply state that the decrease in electronegativity of Sc is what leads to its increased reactivity, without comparing it directly to that of nitrogen. One could do this quantitatively using the reference at hand.

Author reply: We modified the corresponding sentence in the main text: *"It leads to the significant increase of difference in electronegativity between N and Sc above 60 GPa (Supplementary Fig. 3b), which increases the chemical reactivity of scandium, decreases the potential kinetic barriers of reactions, and leads to the appearance of more local minima in the energy landscape."* Also, to show a quantitative change of electronegativities of N and Sc as well as their difference vs pressure we added the additional Supplementary Figure 3.

(2) The authors write: We provided CIF files for all new phases discussed in the paper upon the initial submission, and the quality of experimental structural data can be easily accessed through the CIF files.

Indeed, and my apologies for missing this the first time around. My age is catching up to me. I agree with the authors that these data are of high quality, and stand by my suggestion to include representative images of the raw data, which show extraordinary diffraction quality for DAC work.

Author reply: We appreciate Reviewer #2' high assessment of the quality of our data.